# The Role of Manager Compensation and Integrated Reporting in Company Value: Indonesia vs. Singapore

**St. Dwiarso Utomo \*, Zaky Machmuddah \***  **and Dian Indriana Hapsari**

Faculty of Economic and Business, Universitas Dian Nuswantoro, Jawa Tengah 50131, Indonesia; dian.indriana@dsn.dinus.ac.id
\* Correspondence: dwiarso.utomo@dsn.dinus.ac.id (S.D.U.); zaky.machmuddah@dsn.dinus.ac.id (Z.M.)

**Abstract:** The disclosure of integrated reporting elements can reduce information asymmetry for investors when valuing a company. This study aimed to empirically evaluate the effect of manager compensation, directly or indirectly, on firm value, through the mediating role of the disclosure of integrated reporting elements. The research sample included manufacturing companies listed on the Indonesia Stock Exchange (IDX) and the Singapore Stock Exchange (SGX). The method of analysis was PLS-SEM, using the WarpPLS 7.0 application. The results showed that compensation significantly affects firm value and the disclosure of integrated reporting elements. Integrated reporting has a significant positive impact on firm value. In addition, the disclosure of integrated reporting can mediate the impact of manager compensation on increasing firm value. This research theoretically supports agency theory, disclosure theory, and signal theory, although it is not fully applicable to each country or region of the sample company. The current research contributes to the understanding of the importance of a company's integrated reporting disclosure in improving company value among investors. Integrated reporting describes how a company creates value over time. Our results also suggest that regulators should oblige public companies to disclose integrated reporting.

**Keywords:** integrated reporting; compensation; information asymmetry; firm value





## 1. Introduction

The increase in COVID-19 cases and the tight rules introduced in response to the pandemic have caused a contraction in the growth of manufacturing companies. Data from the newest IHS Markit Purchasing Managers' Index (PMI) have indicated a decrease across all manufacturing sectors in the ASEAN as of July 2020 (https://ekonomi.bisnis.com/; accessed on 3 February 2021). Indonesia and Singapore are included among the ASEAN countries. Indonesia has suffered a greater contraction of its PMI index, from 53.5 to 40.1 in July 2020. In Singapore, the manufacturing sector recorded a significant rebound, from 50.0 to 56.3 in July 2020 (https://ekonomi.bisnis.com/; accessed on Wednesday, 3 February 2021). Company management has had a pivotal role in these outcomes.

Modern companies do not separate ownership and control functions (Berle and Means 1932; Jensen and Meckling 1976). Control of the company is left entirely to the manager, carrying out their role as a principal agent. The consequence of this delegation gives managers the essential task of maximizing the company's value, which increases shareholder wealth (Sucuahi and Cambarihan 2016). Compensation equal to the manager's achievements may be provided as an incentive to carry out his duties. The relationship between corporate management compensation and firm value is significant for researchers and practitioners. The settlement provided to managers can attract and motivate capable employees to improve company performance (Larkin et al. 2012).

On the other hand, the separation of ownership and control functions can potentially lead to asymmetric information on the company's condition. Information asymmetry is a condition where managers control the flow of information to owners or shareholders.

Managers who act as agents for shareholders tend to know more about the company's market value (Myers and Majluf 1984). Thus, these shareholders find it difficult to objectively assess the company's quality based on the asymmetry of the information. Shareholders will, on average, rate the value of shares of the company lower than they are in reality, thus harming the firm's value (Fosu et al. 2016).

Signaling theory states that good companies can differentiate themselves from bad companies by sending credible signals about their quality to the capital market (Spence 1973). The disclosure of better-quality reports can affect company value (Lee and Yeo 2015). Recently, a new reporting paradigm has been considered, in which the company's economic, social, and environmental activities are integrated to provide a more holistic view of company performance and ensure that ethical responsibility is at the forefront of business activities (Lodhia 2015). Managers can create company value by implementing holistic reporting, which is known as integrated reporting. The concept of integrated reporting has become increasingly popular over the last few years. In its simplest form, integrated reporting can be understood as the convergence of sustainability reports and financial reports into one "narrative"—a communication aimed primarily at investors. The top management provide their views on how sustainability issues and initiatives are expected to contribute to long-term business growth (Churet and Eccles 2014).

Previous studies have shown that implementing integrated reporting has a positive impact on firm value (Mervelskemper and Streit 2016; Martinez 2016; Barth et al. 2017; Cosma et al. 2018; El Deeb 2019). Obeng et al. (2020) stated that integrated reporting has a positive effect on the quality of accounting information. Lee and Yeo (2015) stated that companies that disclose integrated reporting obtain greater benefits than the costs incurred, and that integrated reporting reduces information asymmetry between insiders and external investors. Cosma et al. (2018) showed that the stock market reacts positively to award announcements for non-financial companies with high-quality integrated reporting. Their study encouraged investment in improving the quality of integrated reporting. However, the research findings of Churet and Eccles (2014), Suttipun (2017), and Nurkumalasari et al. (2019) suggested that integrated reporting does not affect company value, implying that integrated reporting is not needed by stakeholders in ASEAN territories.

Based on the above, managers have the arduous task of maximizing the company's value for the benefit of shareholders, for which they receive compensation incentives from owners (shareholders). There is a positive relationship between manager compensation and firm value (Dah et al. 2012; Basuroy et al. 2014; Page 2018). Managers can decide to invest in implementing integrated reporting disclosures, which can create value for shareholders. However, Garcia-Sanchez et al. (2020) stated that powerful CEOs may refuse to utilize integrated reporting disclosure, and this behavior is not modified by company incentives. In this case, the greater growth opportunity influences the refusal of CEOs to disclose integrated information about value creation as a consequence of its potential utilization by competitors. Furthermore, we are interested in whether compensation for managers also has a positive relationship along with integrated reporting, and in the relationship between manager compensation and firm value. Although research on the relationship between managers' compensation variables and integrated reporting remains scarce, previous studies have shown a positive relationship between the role of manager compensation and the disclosure of reports that resemble the concept of integrated reporting. Al-Shaer and Zaman (2017) demonstrated a positive effect of CEO compensation on sustainability reporting (which is part of integrated reporting). Thus, there is a framework showing that efforts to increase company value through incentives in the form of compensation are in line with managers' efforts to implement quality integrated reporting disclosures, and that the disclosure of quality integrated reporting is expected to provide added value to the company. Li et al. (2018), Javeed and Lefen (2019), and Sheikh (2019) stated that the relationship between CSR and financial performance is positive because of CEO power. The same was found by Raimo et al. (2020) and Nengzih (2019), who stated that there is a positive relationship between institutional ownership

and integrated reporting quality. However, the other research findings of Raimo et al. (2020) showed that concentrated ownership, managerial ownership, and state ownership negatively affect integrated reporting quality.

This study explores the role of manager compensation in increasing firm value by applying integrated reporting elements. Research on integrated reporting has recently received increased attention, and its role in creating corporate value is of significant interest. However, research linking manager compensation and implementing integrated reporting to an increase in firm value remains scarce.

## 2. Literature Review

Agency theory suggests that corporate governance is based on conflicts of interest between owners (shareholders), managers, and the leading debt financing providers. Each group has different interests and goals. These differences in interests and goals lead to agency conflicts or agency problems (Ross 1973; Jensen and Meckling 1976). Jensen and Meckling (1976) argued that when managers act in the interests of shareholders, managers bear all the costs of failing to achieve company goals, and earn little profit. Therefore, incentives must be given to management in order to increase their willingness to make value-maximizing decisions, or decisions that benefit shareholders, namely, by maximizing the value of the owner's shares. Several methods for reducing agency problems have been suggested, including designing remuneration packages for executive directors and senior managers that incentivize them to act in the best interests of shareholders.

Spence (1973) stated that good companies can differentiate themselves from bad companies by sending credible signals about their quality to the capital market. Ross (1977) showed that companies with high debt can signal that the company is more optimistic and of good quality compared to companies with low debt. In addition, signal theory suggests that company insiders generally know more about the company's prospects than external parties. Signal theory is fundamentally concerned with reducing information asymmetry between two parties (Spence 2002). To reduce information asymmetry, managers (insiders) are advised to provide the information needed by investors or potential investors (Dainelli et al. 2013). Companies that offer better information can influence investors' economic decisions, and attract them to contracts with better benefits than other companies that provide lower quality information (Grossman and Stiglitz 1980).

### 2.1. The Effect of Manager Compensation on Firm Value

Agency theory underlies corporate strategies for increasing firm performance and value through compensation policies (Jensen 1986). From a behavioral perspective, compensation is a determinant of employee effectiveness, as an incentive to improve performance. The performance of any company is influenced by many critical managerial decisions, such as how to price goods, which markets to enter, and how to deal with competition. The quality of such decisions depends not only on the manager's ability, but also on the incentives provided to them to create shareholder value (Byrd et al. 1998).

Several empirical studies support the notion that compensation is an incentive for managers to create value for shareholders, including research conducted by Patnaik and Padhi (2012). This study examined the effect of equity-based CEO compensation on firm value. In particular, this study examines the interaction between CEO compensation and the percentage of independent directors, and the interaction between CEO compensation and managerial entrenchment. The research findings showed a positive relationship between CEO compensation and firm value. In addition, this study also showed that the percentage of independent directors has a positive impact on the marginal effect of EBC on solid value.

Feng et al. (2015) examined the effect of executive compensation on company performance, moderated by workforce-oriented CSR. Executive compensation is measured by the total salary of executive managers divided by the number of executive managers, whereas Tobin's Q measures company performance. The research results showed that executive compensation positively affects company performance (Tobin's Q), but workforce-oriented

CSR weakens the relationship. In addition, Page's research (Page 2018) showed that variations in the value of CEO compensation affect shareholder wealth. This study confirmed that agency conflict can reduce shareholder wealth value, so attractive compensation can be an incentive for CEOs to increase shareholder wealth. Based on the theoretical and empirical studies described, the first hypothesis was formulated as follows:

**Hypothesis 1 (H1).** *The manager's compensation influences the value of the firm.*

### 2.2. The Effect of Manager Compensation on the Disclosure of Integrated Reporting Elements

Disclosure theory suggests that voluntary company performance information disclosure reduces information asymmetry (Verrecchia 1983; Dye 1985). The disclosure of a report can be due to mandatory disclosure or voluntary disclosure. Mandatory disclosure covers the disclosures required by the obligations of accounting standards. In contrast, voluntary disclosure is the disclosure of information in excess of accounting standards.

Managers play an essential role in executing the company's strategy based on the governance and executive compensation policies. The goal is to convince investors that corporate governance policies can make a real contribution to strategic decision-making and the company's long-term vision (IFA 2017). Thus, the executive compensation manager can incentivize the manager to carry out quality disclosures, such as integrated reporting.

Some empirical studies support the notion that the role of manager compensation for the disclosure of reports resembles the concept of integrated reporting, such as a study by Al-Shaer and Zaman (2017), which showed a positive effect of CEO compensation on the disclosure of sustainability reporting. Another study, conducted by Callan and Thomas (2014), showed a positive effect of CEO compensation on CSR disclosure activities. In addition, through an empirical analysis, Karim et al. (2018) found that increasing CEO compensation increases CSR disclosure. The second hypothesis was formulated as follows:

**Hypothesis 2 (H2).** *The manager's compensation influences the disclosure of integrated reporting elements.*

### 2.3. The Effect of the Disclosure of Integrated Reporting Elements on Firm Value

Signal theory states that good companies can differentiate themselves from bad companies by sending credible signals about their quality to the capital market (Spence 1973). Signal theory is fundamentally concerned with reducing information asymmetry between two parties (Spence 2002). To reduce information asymmetry, managers (insiders) are advised to provide information needed by investors or potential investors (Dainelli et al. 2013). Thus, the disclosure of integrated reporting elements can be interpreted as a credible signal for companies, which can add value to the company's stock valuation.

Furthermore, several empirical studies support the idea that integrated reporting can positively affect the company's value. In their research, Lee and Yeo (2015) found that integrated reporting has a more substantial positive impact on companies with higher organizational complexity, by increasing information in a complex corporate environment, such as in companies with significant assets, with many business segments, and large investments. Additional analysis showed that firms with a high degree of integrated reporting outperform firms with low reporting, in terms of their stock market and accounting performance.

Martinez (2016) showed that integrated reporting is positively related to market value (Tobin's Q) and expected future cashflow. That study also confirmed that the disclosure of integrated reporting increases investors' perceptions of the company's future cashflow. In addition, Cosma et al. (2018) tested whether an investment in integrated reporting can increase a company's market value. Integrated reporting was proxied by an award given to companies for the implementation of quality IR. The study found that the stock market reacted positively to the announcement of IR quality awards in non-financial companies.

The results of this study should encourage managers to invest in improving the quality of IR disclosures. The third hypothesis was formulated as follows:

**Hypothesis 3 (H3).** *The disclosure of integrated reporting elements influences the value of the firm.*

*2.4. Integrated Reporting Mediates the Effect of Manager Compensation on Firm Value*

Manager compensation can increase managers' efforts to invest in quality integrated reporting disclosures, whereas quality IR disclosures can increase the company's market value. Therefore, there is a framework that links integrated reporting to the relationship between manager compensation and firm value. In addition, research by Shim and Kim (2015) showed that CEO compensation in the pre-SOX period greatly determines market-based performance (strong value). The results of this study confirmed the impact of the SOX Act, suggesting that more robust internal control systems and reliable financial reporting are required to encourage CEOs to maximize shareholder value. The framework of the relationship between CEO compensation and firm value requires more robust internal control mediation and a reliable financial reporting system. A reliable financial reporting system can be a proxy in the context of implementing integrated reporting. Thus, a fourth hypothesis was formulated as follows:

**Hypothesis 4 (H4).** *The disclosure of integrated reporting elements mediates the effect of manager compensation on firm value.*

Based on the development of the hypotheses based on the previous theoretical and empirical studies, a practical research model was developed, and is presented in Figure 1.

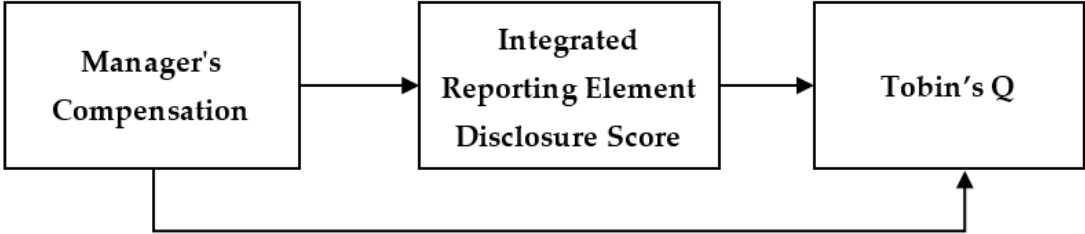

**Figure 1.** Empirical research model.

### 3. Method

The sample consisted of 30 companies from Indonesia and 30 companies from Singapore, comprising a total of 60 companies engaged in the manufacturing sector. This sub-sector of the business sector is quite diverse, including the consumer goods industry, the essential chemicals industry, and various other industries. This study covered a sample period of five years, from 2016 to 2020. The panel data amounted to 300 observation from 60 manufacturing companies, multiplied by the five years of the research period. The data were secondary data published by each company in financial reports, annual reports, and sustainability reports.

All variables in this study were manifest variables, which are observed variables with formative constructs. The measurement of the research variables is presented in Table 1.

In this study, we analyzed partial least squares (PLS) structural equation modeling (SEM), using the WarpPLS version 7.0 program to test the hypothesis.

**Table 1.** Measurement of the research variables.

| Variable | Variable Measurement | Reference Source |
|---|---|---|
| Cash-based manager compensation (independent variable) | Natural logarithm of the annual salary plus bonuses earned in the fiscal year | (Shim and Kim 2015; Al-Shaer and Zaman 2017) |
| Integrated reporting element disclosure score (mediation variable) | The integrated reporting index element disclosure score is based on items/indicators from (1) an overview of the organization and the external environment; (2) governance; (3) business models; (4) risks and opportunities; (5) strategy and resource allocation; (6) performance; (7) outlook; and (8) essential preparation and presentation. The integrated reporting index is calculated by dividing the total items disclosed by the total items disclosed (8 items). | (Lee and Yeo 2015) |
| Tobin's Q (dependent variable) | Tobin's Q = (VMS + D)/TA where: <br> - VMS = market value of all outstanding shares, i.e., company share price × extraordinary shares. <br> - TA = Company assets, such as; cash, accounts receivable, inventory, and a book value of the land. <br> - D = debt. | (Lindenberg and Ross 1981) |
| Company size (control variable) | Natural logarithm of total assets. | (Basuroy et al. 2014; Feng et al. 2015) |
| Leverage (control variable) | Total debt divided by total assets. | (Desoky and Mousa 2013; Lahouel et al. 2014; Zou et al. 2015). |

Source: several empirical research results developed for this study.

## 4. Results and Discussion

Before evaluating the relationships between variables, we first assessed the goodness of fit of this research model. The goodness-of-fit results for the combined panel data from Indonesian and Singaporean manufacturing companies are shown in Table 2.

**Table 2.** Goodness-of-fit structural model.

| Criteria | Parameter | Rule of Thumb |
|---|---|---|
| Average path coefficient (APC) | 0.156, $p < 0.01$ | acceptable $p < 0.05$ |
| Average block VIF (AVIF) | 1.041 | acceptable if $\leq 5$, ideally $\leq 3.3$ |
| Average full collinearity VIF (AFVIF) | 1.339 | acceptable if $\leq 5$, ideally $\leq 3.3$ |
| Tenenhaus GoF (GoF) | 0.485 | small $\geq 0.1$, medium $\geq 0.25$, large $\geq 0.36$ |
| Simpson's paradox ratio (SPR) | 0.714 | acceptable if $\geq 0.7$, ideally = 1 |
| R-squared contribution ratio (RSCR) | 0.996 | acceptable if $\geq 0.9$, ideally = 1 |
| Statistical suppression ratio (SSR) | 0.857 | acceptable if $\geq 0.7$ |

Source: secondary data.

Based on Table 2, this research model was a good fit; the *p*-value for APC < 0.05, with the APC value = 0.156. Likewise, the resulting AVIF and AFVIF values were relatively small, at (<) 3.3, meaning there was no multicollinearity between the indicators and between exogenous variables. The resulting GoF was 0.485 > 0.36, which means that the fit of the model was perfect. For SPR, RSCR, and SSR, the values were above the required values, indicating no causality problem in the model.

The results of the relationship estimations between variables were presented based on three data panels, namely, the data panels of the Singaporean manufacturing companies, the Indonesian manufacturing companies, and the combined data of the Singaporean and Indonesian manufacturing companies. Each data panel was presented in two models: a model without control variables and a complete model with control variables. Model 1 is the Singapore panel data non-control variable. Model 2 is the Singaporean data panel with control variables. Model 3 is the panel of Indonesian data with non-control variables. Model 4 is the Indonesian data panel with control variables. Model 5 is the combined non-variable control panel. Model 6 is a combined panel with control variables. The estimation model of the relationship between variables is presented in Table 3.

**Table 3.** Results of the estimation of the relationships between variables.

| | Singapore | | Indonesia | | Combined | |
| --- | --- | --- | --- | --- | --- | --- |
| **Description Path** | **Model 1** | **Model 2** | **Model 3** | **Model 4** | **Model 5** | **Model 6** |
| Comp $\rightarrow$ Q | −0.104 *** | 0.063 * | 0.126 *** | 0.171 ** | −0.001 | 0.202 ** |
| Comp $\rightarrow$ IRR | 0.281 *** | 0.170 * | 0.380 *** | 0.313 *** | 0.208 *** | 0.397 ** |
| IRR $\rightarrow$ Q | 0.089 | 0.110 * | 0.346 *** | 0.351 *** | 0.135 ** | 0.142 *** |
| Comp $\rightarrow$ IRR $\rightarrow$ Q | 0.025 | 0.019 | 0.132 *** | 0.110 ** | 0.028 ** | 0.056 * |
| Control Variable | | | | | | |
| SIZE $\rightarrow$ Q | - | -0.153 * | - | 0.017 | - | -0.163 * |
| LEV $\rightarrow$ Q | - | 0.670 *** | - | 0.158 *** | - | 0.640 *** |
| SIZE $\rightarrow$ IRR | - | 0.148 * | - | 0.080 | - | −0.367 ** |
| LEV $\rightarrow$ IRR | - | 0.003 | - | −0.046 | - | −0.033 * |
| SIZE $\rightarrow$ IRR $\rightarrow$ Q | - | 0.016 | - | 0.028 | - | −0.052 * |
| LEV $\rightarrow$ IRR $\rightarrow$ Q | - | 0.000 | - | −0.016 | - | −0.005 |

***, **, and * denote significance levels at 0.001, 0.05, and 0.1, respectively. Source: secondary data.

The estimation results can also be presented in a path analysis diagram. However, the path analysis diagram shown is only for the combined panel data, especially Model 6 (the model uses control variables), considering that Model 6 was the primary model used in this study. The path analysis diagram is presented in Figure 2.

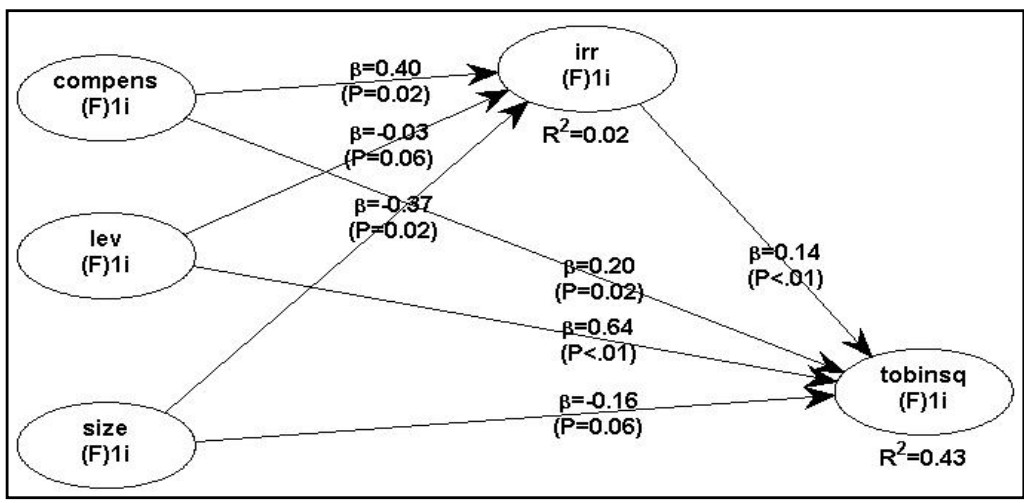

**Figure 2.** Path analysis chart.

Based on Figure 2, the R-squared ($R^2$) value for variations that affect integrated reporting (IRR) is 0.02, which means that the effect of compensation variations and the control variable leverage and size on integrated reporting (IRR) variations is 2%, and the remaining 98% is affected. Other variables were outside the scope of this research model. The R-squared ($R^2$) value for variations that affect Tobin's Q was 0.43, which means that

the effect of variations in compensation, integrated reporting, and control variables of leverage and size on Tobin's Q variations is 43%, and other variables outside the scope of this research model influence the remaining 57%. The results indicated that when the compensation of the CEO is tested in relation to IRR disclosure using the control variables of leverage and size, the R-squared ($R^2$) value is very low, at just 2%. However, when the compensation of the CEO is tested with Tobin's Q and is mediated by integrated reporting disclosure, the R-squared ($R^2$) value is larger, at 43%. This means that integrated reporting disclosure plays a role as a mediating variable in the relationship.

### 4.1. The Effect of Manager Compensation on Firm Value

The results of the estimation of the primary model in this study indicated that manager compensation positively affects firm value. This means that an increase or decrease in manager compensation will have a positive or negative impact on the company's value. The results of this study support agency theory, which states that the company's strategy increases the performance and weight of the company through compensation policies (Jensen 1986). This study also supports the findings of previous studies which stated that compensation is an incentive for managers to create value for shareholders (Patnaik and Padhi 2012; Feng et al. 2015; Page 2018).

There is a slight difference in the effect of compensation on firm value for manufacturing companies in Indonesia and Singapore. In Indonesia, both the size of the companies and leverage have the same effect as increasing manager compensation on encouraging agents to act to increasing shareholder interests. Meanwhile, in Singapore, the impact of payment on firm value is controlled by the size and leverage of the firm. This means that payment has a positive impact on Tobin's Q, depending on the size and influence of the company.

The results showed that if the model does not include compensation control variables, firm value is harmed. In this case, agency theory does not fully apply to manufacturing companies in Singapore. This result can be interpreted as representing lower compensation but higher firm value for Singaporean manufacturing companies. Managers continue to work hard to increase the company's value as the organization's primary goal. This motive aligns with stewardship theory, which states that management is not motivated by individual pursuits but rather is aimed at the main outcome goals for the organization's benefit (Davis et al. 1997). This is supported by descriptive statistical data, which show that the average value of Tobin's Q in Singapore is higher than in Indonesia. When selection of the firm is controlled for size and leverage, agency theory becomes relevant in the relationship between compensation and firm value.

### 4.2. The Effect of Manager Compensation on the Disclosure of Integrated Reporting Elements

The estimation results of the research showed that manager compensation has a positive effect on the disclosure of integrated reporting elements. The results of this study can be interpreted to mean that an increased payment to managers encourages the disclosure of integrated reporting elements. These results support disclosure theory, which attempts to explain why companies voluntarily disclose information related to company performance (Verrecchia 1983; Dye 1985). The study results are also in line with previous studies that have stated that executive managers who receive compensation are incentivized to carry out quality disclosures, such as integrated reporting (Callan and Thomas 2014; Al-Shaer and Zaman 2017; Karim et al. 2018).

The estimation results showed that manufacturing companies in both Indonesia and Singapore disclose integrated reporting elements to reduce the occurrence of asymmetric information for shareholders regarding the actual condition of the company. Thus, the stakeholders share the same perception when evaluating the company's overall performance, in terms of economic, social, and environmental performance.

*4.3. The Effect of the Disclosure of Integrated Reporting Elements on Company Value*

The estimation results of the research showed that the disclosure of integrated reporting elements has a positive effect on firm value. The study results imply that increased exposure of the company's integrated reporting elements will positively impact investors, so the market will rate the company more highly. The research results align with signaling theory, which states that good companies can differentiate themselves from bad companies by sending credible signals about their quality to the capital market (Spence 1973). These results also support previous empirical studies which concluded that greater disclosure of integrated reporting can positively affect firm value (Lee and Yeo 2015; Martinez 2016; Cosma et al. 2018).

The estimation results of the research model as a whole gave the same results for Model 1 only in the Singapore data panel, which did not involve control variables that produce different estimates; this suggested that the disclosure of integrated reporting elements does not affect firm value. This result can be interpreted to mean that the value of manufacturing companies in Singapore may only be positively influenced by the disclosure of IRR in companies depending on size and leverage. Meanwhile, for companies in Indonesia, the effect of IRR disclosure on firm value does not differ based on company size and leverage.

*4.4. Integrated Reporting Mediates the Effect of Manager Compensation on Firm Value*

The estimation results of the primary research model showed that the disclosure of integrated reporting can mediate the effect of manager compensation on firm value. That is, increasing manager compensation encourages management to carry out more quality disclosure of integrated reporting elements that impact firm value. Empirically, this research is in line with the analysis of Shim and Kim (2015). They found that there is a framework for the relationship between CEO compensation and firm value, requiring more robust internal control mediation and a reliable financial reporting system.

The results of the overall research model showed that there are differences between manufacturing companies in Indonesia and Singapore. In the Singapore data panel, both Model 1 (non-control variables) and Model 2 (with control variables) produced estimates showing that IRR does not mediate the effect of manager compensation on firm value. Meanwhile, in the Indonesian data panel, both Model 3 (non-control variables) and Model 4 (with control variables) indicated that IRR can mediate the effect of manager compensation on firm value. Based on these results, the effect of the disclosure of IRR on increasing the value of manufacturing companies in Singapore only acts as an independent variable. For manufacturing companies in Indonesia, the disclosure of IRR serves as an independent variable and also acts as a mediating variable in increasing firm value. This means that IRR in manufacturing companies in Singapore does not mediate the relationship between the compensation of the CEO and company value. Nevertheless, the role of IRR disclosure in manufacturing companies in Singapore can directly increase company value. For both Indonesian manufacturing companies and Singaporean manufacturing companies (see Figure 2 and Model 6), IRR disclosure mediated the effect of the compensation of managers on the company value.

**5. Conclusions**

The study results showed that manager compensation has a positive effect on firm value. Increasing payments can be an incentive to increase firm value. These results applied to manufacturing companies in both Indonesia and Singapore. However, for manufacturing companies in Singapore, the positive effect of compensation on firm value depended on the size and leverage of the company. The results of this study are in line with agency theory, which suggests that firm value can be increased through compensation policies (Jensen 1986). However, for manufacturing companies in Singapore, agency theory is not fully applicable, because the role of compensation in increasing the company's value depends on the size and leverage of the company.

Manager compensation has a positive impact on the disclosure of elements of integrated reporting. Increasing wages encourages managers to increase the disclosure of elements of integrated reporting that are of higher quality. This applied equally to manufacturing companies in Indonesia and Singapore. These results align with and support disclosure theory, which suggests that companies voluntarily disclose information on company conditions to reduce information asymmetry for investors (Verrecchia 1983; Dye 1985).

Other research results have demonstrated that the disclosure of integrated reporting elements has a positive effect on firm value. The higher the quality of the company's integrated reporting elements, the more it can increase investors' valuations. Signal theory suggest that companies can communicate their value to investors by giving a positive signal to the market (Spence 1973). However, signal theory does not fully apply to manufacturing companies in Singapore, because the positive effect of the disclosure of performance reports on firm value depends on the size and leverage of the company.

Furthermore, the results of other studies showing the mediating role of the disclosure of integrated reporting elements on the effect of manager compensation in increasing firm value are fully applicable to manufacturing companies in Indonesia. For manufacturing companies in Singapore, the disclosure of integrated reporting elements does not play a role as a mediating variable, though it is an independent variable.

This study has several limitations. The sample only considered the manufacturing sectors of Indonesia and Singapore. The business sector and other Asian countries were not included. Future research could expand the model by using other sectors, such as the mining, plantation, and banking industries, and other Asian countries, so that the results can be generalized. The current study confirms the importance of integrated reporting disclosure to improve company value for investors or investor candidates. Integrated reporting describes how a company creates value over time. The results of this study also contribute support for the introduction of regulations to oblige public companies to disclose integrated reporting.

**Author Contributions:** Conceptualization, S.D.U.; methodology, D.I.H.; software, Z.M.; validation, S.D.U.; formal analysis, S.D.U. and Z.M.; investigation, D.I.H. and Z.M.; resources, S.D.U. and Z.M.; data curation, D.I.H.; writing—original draft preparation, Z.M.; writing—review and editing, S.D.U.; visualization, S.D.U.; supervision, S.D.U.; project administration, D.I.H. All authors have read and agreed to the published version of the manuscript.

**Funding:** This research was funded by the Ministry of Research and Technology/National Research and Innovation Agency of the Republic of Indonesia, grant number 1868/E4/AK.04/2021.

**Acknowledgments:** We would like thank the Ministry of Research and Technology/National Research and Innovation Agency of the Republic of Indonesia and Universitas Dian Nuswantoro.

**Conflicts of Interest:** The authors declare that they have no conflict of interests regarding this article.

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
