# Peer review of "The Role of Manager Compensation and Integrated Reporting in Company Value: Indonesia vs. Singapore"

_economies, doi:10.3390/economies9040142_

Round 1
Reviewer 1 Report
Please find attached the Review Report.

Author Response
Thank you very much for your suggestions and criticism. We have improved our articles according to your suggestions and criticism to make our articles better. The suggestions and criticism you provide are very useful to make our articles better.

Reviewer 2 Report
The paper investigated the effect of manager's compensation directly or indirectly on firm value through the mediating role of disclosing integrated reporting elements on a sample of listed companies listed on the Indonesia Stock Exchange (IDX) and the Singapore Stock Exchange (SGX).
The paper is intersting, but there are some aspects to need improved.
The literature review must be update with recent articles from lat two years. After a quick search I found the articles below, but you can also find other suitable articles.
Garcia-Sanchez, I. M., Raimo, N., & Vitolla, F. (2020). CEO power and integrated reporting. Meditari Accountancy Research. Vol. 29 No. 4, pp. 908-942.
Nengzih, N. (2019). The Role of Corporate Governance to Integrated Reporting (Survey on Indonesia’s State-Owned Enterprises/SOEs), Saudi Journal of Economics and Finance, p. 314 https://www.saudijournals.com/media/articles/SJEF-38-314-322-c.pdf
Nurkumalasari, I. S., Restuningdiah, N., & Sidharta, E. A. (2019). Integrated reporting disclosure and its impact on firm value: Evidence in Asia. International Journal of Business, Economics and Law, 18(5), 99-108.
Obeng, V. A., Ahmed, K., & Miglani, S. (2020). Integrated reporting and earnings quality: The moderating effect of agency costs. Pacific-Basin Finance Journal, 60, 101285.
Raimo, N., Vitolla, F., Marrone, A., & Rubino, M. (2020). The role of ownership structure in integrated reporting policies. Business Strategy and the Environment, 29(6), 2238-2250.
The hypotheses could be reformulated, eg for H1:
The manager's compensation influence the firm value or
There is a direct positive correlation between the two variables.
It is necessary to explain in the paper which is the model used for H2, I did not understand which is the dependent variable, however the model is not validated if R2 is 0.02 as it turns out from line 275.
The explanations in the lines 274-280 are a bit ambiguous, to be checked in English as well.
I would also like to see a well-developed discussion comparing and contrasting results presented in the paper with existing papers and then a contributions to literature. Also how the results can be generalized to companies from other countries.
Author Response
Thank you so much for your criticism. We have improved our articles according to your suggestions to make our articles better. The suggestions you provide are very useful to make our articles better.

Reviewer 3 Report
- This is a well-written study, however, which central idea is somewhat simplistic. It assumes that managerial compensation has a direct effect on firm value. I accept the significant correlation shown between the two, however, the authors should thoroughly point out in the discussion section that the relationship is not necessarily direct. Possible factors that could play a mediating role in this regard should be analyzed, at least on a theoretical level. Examples include competent leadership, organizational culture, motivational efforts, and related management systems, and so on.
- Although the individual hypotheses have been derived logically, their substantiation in the literature is a bit weak, their narrative needs to be made more robust.
- In part, it is worthwhile to address the limitations of the research. For example, examining managerial compensation alone is not necessarily informative, instead it would be worthwhile to test the strength of the relationships by measuring satisfaction with managerial compensation — as expectations, and thus satisfaction, can vary widely from individual to individual.
Author Response

(The authors gave the same response as above.)

Round 2
Reviewer 1 Report
Dear Author(s),
The revised manuscript titled “THE ROLE OF COMPENSATION MANAGER AND APPLI-CATION INTEGRATED REPORTING ON COMPANY VAL-UE: INDONESIA VS SINGAPORE” (Manuscript ID: economies-1361586) improved in a proper manner. The author(s) incorporated the suggestions and recommendations as formulated throughout the prior review round. As well, the author(s) provided suitable arguments and replied properly to each concern formulated priorly. As such, the quality of the submission enhanced significantly. Therefore, I recommend paper acceptance to publication.
Reviewer 2 Report
The paper is improved with my recommendations.